# Evaluating Drug Interaction Risks: Nirmatrelvir & Ritonavir Combination (PAXLOVID^®^) with Concomitant Medications in Real-World Clinical Settings

**DOI:** 10.3390/pathogens13121055

**Published:** 2024-11-30

**Authors:** Petra Zatovkaňuková, Dalibor Veselý, Jiří Slíva

**Affiliations:** Department of Pharmacology, Third Faculty of Medicine, Charles University, Ruská 87, 100 00 Prague, Czech Republic; dalibor.vesely@lf3.cuni.cz (D.V.); jiri.sliva@lf3.cuni.cz (J.S.)

**Keywords:** COVID-19, drug interactions, nirmatrelvir, Paxlovid, ritonavir

## Abstract

Background: This research article delves into the battle against the COVID-19 pandemic, focusing on the efficacy and, particularly, the safety of the combination of nirmatrelvir with ritonavir, which is found in the pharmaceutical product Paxlovid^®^. This study aims to analyze the potential interactions of commonly prescribed medicinal products with Paxlovid^®^, shedding light on its utilization in specific medical fields. Methods: Prescription data from the Czech Republic’s Institute of Health Information and Statistics (IHIS CR) was analyzed, covering 4 million COVID-19 patients and 87.5 million medication records from September 2019 to February 2022. This study focused on potential drug interactions with Paxlovid among the 50 most frequently prescribed medications, with particular attention to four specialties: general medicine, internal medicine, infectious diseases, and diabetology. Results: In this study of the 50 most commonly prescribed drugs, 56% showed no interaction with Paxlovid, 30% had a potential for interaction, and 14% were not specifically mentioned in relation to Paxlovid, with no drugs found to be contraindicated overall. However, in specific medical fields, including diabetology, infectious diseases, internal medicine, and general medicine, certain drugs had potential interactions when co-administered with Paxlovid. Conclusions: Paxlovid remains a valuable option for early COVID-19 treatment but requires a careful consideration of potential drug interactions, especially in high-risk specialties. A thorough assessment of concurrent medications is essential to optimize safety and efficacy in patients receiving Paxlovid.

## 1. Introduction

Coronavirus disease (COVID-19) is an infectious disease caused by the SARS-CoV-2 virus. It sparked a global pandemic that started in March 2020 and lasted for three years. According to the published data from the World Health Organization (WHO), as of 18 October 2023, there have been 771,407,825 confirmed cases of COVID-19, resulting in 6,972,152 reported deaths worldwide [1].

SARS-CoV-2 contains four key structural proteins: spike glycoproteins (S), small envelope glycoproteins (E), membrane glycoproteins (M), and nucleocapsid (N), alongside several accessory proteins [2]. Among these, the spike (S) protein is critical for enabling the virus to bind to the angiotensin-converting enzyme 2 (ACE2) receptor in human cells, facilitating viral entry. This protein quickly became a focus for vaccine development to help prevent infection [3].

In addition to vaccines, various drugs have been investigated or approved to combat COVID-19. While many, including antimalarials and antiparasitics, proved ineffective, several have shown promise. These include remdesivir, favipiravir, molnupiravir, monoclonal antibodies like bamlanivimab, and, most notably, Paxlovid^®^, which combines nirmatrelvir and ritonavir.

Nirmatrelvir (PF-07321332) is an oral antiviral that targets the SARS-CoV-2 3-chymotrypsin-like cysteine protease (Mpro) enzyme, essential for viral replication [4]. The selectivity of Mpro minimizes the risk of off-target effects in human cells [5]. Nirmatrelvir has demonstrated strong inhibition of Mpro and a significant reduction in viral replication in vitro across various coronaviruses. In animal models, the oral administration of nirmatrelvir significantly decreased viral loads in the lungs compared to the placebo [5].

Nirmatrelvir’s primary metabolic pathway is through the cytochrome P450 3A4 enzyme [4]. In contrast, ritonavir is primarily recognized as a protease inhibitor targeting this particular isoform. Ritonarvir was originally approved for HIV treatment. While it possesses limited intrinsic antiviral activity, its principal function is to enhance the efficacy of other protease inhibitors. Beyond its initial use against HIV protease, studies have revealed its capacity to inhibit cytochrome P450-3A4.

Consequently, co-administering Nirmatrelvir with a low dose (100 mg) of ritonavir increases the systemic exposure of Nirmatrelvir, thereby supporting its therapeutic effect. The first study in healthy volunteers confirmed the favorable safety profile of this combination, even at the highest dose and exposure evaluated (500 mg nirmatrelvir plus 100 mg ritonavir, twice daily for 10 days).

Based on these findings, a double-blind phase II/III EPIC-HR clinical trial was conducted in 2021, with the objective of evaluating the therapeutic impact of this specific combination in symptomatically ill, unvaccinated adults who were not hospitalized but were at a high risk of progressing to a severe form of COVID-19 [6].

The provided data vividly demonstrated that the treatment with nirmatrelvir in combination with ritonavir in the early stages of COVID-19 can fundamentally suppress the progression to severe illness and, at the same time, rapidly reduce the viral load of SARS-CoV-2. Nevertheless, there is significant apprehension in clinical settings regarding its safety due to the potential for drug interactions, chiefly stemming from ritonavir’s inhibitory impact on cytochrome P450.

Based on the results of the clinical studies conducted and a comprehensive evaluation of the therapeutic effectiveness of different antiviral medications, the Czech Ministry of Health issued an Interdisciplinary Opinion in July 2022 concerning the utilization of antivirals for the treatment and prevention of COVID-19 progression. The opinion highlights a decrease in the relative risk of hospitalization: 30% with molnupiravir, 87% with remdesivir, and 89% with nirmatrelvir [7].

The pharmacokinetic interactions of Paxlovid^®^ can result due to various mechanisms; however, for the fixed-dose combination (FDC) of nirmatrelvir and ritonavir, these interactions primarily occur at the level of the cytochrome P450 enzyme system. Resources such as the Liverpool database categorize these interactions based on their clinical significance rather than their specific mechanism, incorporating all currently known drug interactions.

Regarding protein binding, nirmatrelvir exhibits species-dependent plasma protein binding, influenced by molecular differences in albumin and alpha-1 acid glycoprotein, which affect binding affinity. Studies in rats, monkeys, and humans show moderate protein binding, with unbound fractions ranging from 0.310 to 0.478 [8]. According to the manufacturer, nirmatrelvir’s protein binding in human plasma is approximately 69%. Ritonavir, on the other hand, shows high plasma protein binding, ranging from 96% to 99.5% in animal studies, and is nonsaturable in humans at concentrations up to 30 µg/mL. Partitioning into blood cells is minimal, and the manufacturer reports its protein binding in human plasma to be approximately 98% to 99% [9].

The aim of our study is to analyze data regarding the utilization and prescription of commonly administered medicinal products in specific medical fields. Our objective is to delineate the prevalent concurrent therapies among patients diagnosed with COVID-19 and the possible drug interactions with Paxlovid.

## 2. Materials and Methods

### 2.1. Data Sources

The analysis primarily relies on data from the registers of the Institute of Health Information and Statistics of the Czech Republic (IHIS CR). The IHIS CR is the organizational unit of the state delegated by the Ministry of Health of the Czech Republic and cooperates with associations of hospitals, expert medical societies, associations of physicians, health insurance companies, and other organizations [10].

The analysis included all patients with a positive test by 22 September 2022 or with a reported diagnosis of COVID-19 according to ICD-10 by 29 April 2022. This encompassed a total of 4,078,065 patients. The analysis focused on all recorded reimbursed medications for this entire study population, covering the period from September 2019 to February 2022. This includes all reimbursed prescription medicines and medicines reported as separately billed medicines. The data exclude drugs that are not separately reported to payers, i.e., primarily drugs used for inpatient admissions that are covered under the flat-rate drug benefit. The dataset under examination contained 87,510,512 records (specific prescriptions or medication administrations) and 119,214,570 medication packages.

For the purpose of identifying medical specialties where frequent prescribing of the medication Paxlovid^®^ is anticipated, data from five health insurance companies (VZP, ČPZP, OZP, RBP, and VoZP), representing approximately 87% of insured individuals in the Czech Republic, were used. These data cover the period from December 2021 to September 2022.

### 2.2. Analysis

In the initial phase, the most commonly prescribed medicinal substances (or ATC [Anatomical Therapeutic Chemical Classification] groups for a given medicine) were identified according to the number of prescriptions and, additionally, the number of packages and the number of patients with at least one record for a given medicinal substance. For the subsequent pharmacological assessment of potential drug interactions with Paxlovid^®^, we focused on the 50 ATC groups with the highest prescription rates. These particular medicinal substances were of utmost importance due to their prevalence among COVID-19 patients, with the possibility of concurrent prescription and potential drug interactions. It is important to highlight that, for this analysis, a more accurate metric than the number of prescriptions would be the count of Usual Daily Therapeutic Doses (UDTDs) per package. However, this approach proved challenging due to the lack of differentiation in the SÚKL (State Institute for Drug Control) codes in the initial dataset from the IHIS.

In the subsequent step, the common medicinal substances (or ATC groups for a given medicine) were identified based on the medical specialties, including general practitioners, internal medicine, infectious diseases, and diabetology. The first three named specialties were selected due to their highest absolute prescription of molnupiravir. A similar distribution among specialties is anticipated for Paxlovid, as indicated by health insurance data up to January 2023. Diabetology, as the fourth specialty, was chosen due to the notable concentration of patients at a high risk of severe COVID-19 outcomes [11]. Additionally, for each of these specialties, we included the top five medicinal substances with the highest prescription rates that were not part of the initial 50 medicinal substances. The most frequently prescribed medicinal substances were recalibrated per physician within their respective specialties in the Czech Republic.

This comprehensive approach resulted in a total of 70 medicinal substances (or ATC groups for a given medicine), each of which underwent a thorough pharmacological evaluation regarding potential drug–drug interactions with the pharmaceutical product Paxlovid^®^. Each group underwent a review of data from two primary sources: the Liverpool checker (referred to as “LC”) [12] and the Summary of Product Characteristics (SPC) for the medicinal product Paxlovid^®^ [13]. In the Liverpool checker, the 5-day nirmatrelvir/ritonavir treatment option was selected, following the current treatment recommendations [9]. The potential interaction mechanisms for each medicinal substance/ATC group were described, along with the recommended actions per the SPC, possible courses of action according to the LC, and recommendations from the Society of Infectious Diseases. An overall risk assessment was also conducted, taking into account a conservative approach that prioritized the higher risk of interaction per the LC or SPC. The overall risk was classified into one of the following categories:➢No interaction;➢Potential weak interaction;➢Potential interaction;➢Contraindication;➢Not mentioned (given the nature of the sources, a similar interpretation to “No interaction” is plausible).

## 3. Results

In contrast to the standard information provided in the Summary of Product Characteristics (SPC), the Liverpool checker offers the possibility of scaling the evidence and intensity, or clinical severity, of potential drug interactions. Table 1 lists the 50 most commonly prescribed medicinal substances (or ATC groups for a given medicine) prescribed across all medical specialties and the potential health risks when combined with the medicinal product Paxlovid^®^. Although there are slight variations between these two sources, they are not considered clinically significant. Table 1 also includes the current recommendations from the SPC, LC, and the Society of Infectious Diseases. Table 2 presents similar data for the four mentioned medical specialties, specifically for the top five most frequently prescribed medicinal substances in each of them.

## 4. Discussion

The majority of potential drug interactions associated with the combination of nirmatrelvir and ritonavir have a pharmacokinetic nature. Nirmatrelvir and ritonavir are substrates of CYP3A, meaning that the plasma levels of these drugs depend directly on the activity of this microsomal system. Induction leads to decreased levels, while inhibition results in increased levels. Furthermore, ritonavir itself acts as a potent inhibitor of CYP 3A4, 2D6, and P-glycoprotein (P-gp). Conversely, it acts as an inducer for CYP 1A2, 2C8, 2C9, and 2C19. Nirmatrelvir can inhibit MDR1, MATE1, OCT1, and OATP1B1 at clinically relevant concentrations [13].

The observed drug interactions in this study are primarily attributed to ritonavir, whose inhibitory effect on CYP3A4 is critical for achieving therapeutic levels of nirmatrelvir. Given that CYP3A4 is the predominant isoform involved in metabolizing various drugs and xenobiotics, the frequency of potential interactions is unsurprising. However, in regular clinical practice, these interactions mainly concern a limited number of co-administered medications; thus, concerns should be assessed judiciously without exaggeration.

CYP3A4 polymorphisms, influenced by intronic SNPs affecting expression levels, categorize individuals as poor, normal, or rapid metabolizers, with most falling into the extensive metabolizer category based on a natural Gaussian distribution. Although CYP3A4 variability is significant, its clinical impact on drug metabolism is generally less pronounced compared to the highly polymorphic CYP2D6 or CYP2C19 isoforms.

Among the 50 most commonly prescribed drugs, 56% have no interactions with Paxlovid. About 30% of these drugs present a potential for interaction, while 14% are not specifically mentioned in relation to Paxlovid. No drugs were found to be contraindicated or to have a weak interaction potential.

In terms of specific medications within the specialties of diabetology, infectious diseases, internal medicine, and general medicine, it is important to highlight that rivaroxaban is contraindicated with Paxlovid. Additionally, there is a potential for interaction with drugs such as apixaban, warfarin, and indapamide when used concurrently with Paxlovid.

Nevertheless, we would like to draw attention to recent news from the European Medicines Agency’s safety committee (PRAC) regarding the concerning risk of serious and potentially fatal adverse reactions associated with Paxlovid^®^ when used alongside specific immunosuppressants. These immunosuppressants, known as calcineurin inhibitors (tacrolimus, ciclosporin, voclosporin) and mTOR inhibitors (everolimus, sirolimus), have been highlighted as posing significant risks when combined with Paxlovid^®^.

In this case, healthcare professionals are advised to carefully assess the potential benefits of Paxlovid^®^ treatment against the heightened risks of serious adverse reactions when used concurrently with these immunosuppressants. In some instances, rapid increases in blood levels of these immunosuppressants have led to toxic levels, resulting in life-threatening conditions [14].

Paxlovid is contraindicated in patients with severe hepatic or renal impairment. No dose adjustment is required for patients with mild or moderate hepatic impairment, or for those with mild renal impairment. However, in patients with moderate renal impairment, the dose should be reduced to nirmatrelvir/ritonavir 150 mg/100 mg every 12 h for 5 days to prevent overexposure [9].

## Figures and Tables

**Table 1 pathogens-13-01055-t001:** Fifty of the most frequently prescribed medicinal substances across all specialties in relation to possible health risks when combined with the medicinal product Paxlovid^®^.

Chemical Substance/ATC Group	Number of Prescriptions *	SPC	Liverpool Checker (LC)	Total Risk	Evidence	Mechanism of Interaction	Recommended Measures According to SPC	Possible Measures According to LC	Recommendation of the Czech Society of Infectious Diseases	Comments
Levothyroxine sodium	2,288,496	possible interaction	no interaction	no interaction	very low	-	monitoring of the TSH level recommended in the first month after starting the treatment	-	-	-
Amides	1,762,441	not mentioned	potential interaction	potential interaction	very low	possible increase in concentration due to inhibition of its metabolism at the level of CYP 3A4	-	simultaneous administration with increased caution	-	lidocaine and bupivacaine mentioned in LC
Atorvastatin	1,742,850	possible interaction	potential interaction	potential interaction	very low	increase in atorvastatin level due to inhibition of CYP 3A4 and P-gp	reducing the dose of atorvastatin to the lowest possible level	due to the short duration of the treatment, it is recommended to suspend statin therapy with resumption of treatment 3 days after stopping Paxlovide treatment; if statin therapy is continued, a reduction in the dose of atorvastatin is recommended	suspend the use	
Amoxicillin and Beta-Lactamase inhibitor	1,724,958	not mentioned	no interaction	no interaction	very low	-	-	-	-	-
Crystalloids	1,488,381	not mentioned	-	not mentioned (no interaction)	-	-	-	-	-	they are not mentioned in the LC; apparently without significant risk of DI
Omeprazole	1,445,928	not mentioned	no interaction	no interaction	very low	-	-	-	-	-
Acetyl salicylic	1,339,123	not mentioned	no interaction	no interaction	very low	-	-	-	-	-
Metformin	1,276,405	not mentioned	no interaction	no interaction	very low	-	-	-	-	-
Rosuvastatin	1,234,425	possible interaction	potential interaction	potential interaction	very low	inhibition of the transporter for rosuvastatin with a subsequent increase in its concentration	administration of rosuvastatin in the lowest possible doses	temporary dose reduction to 10 mg/day or complete withdrawal	temporarily suspend the use	-
Diclofenac	1,217,276	not mentioned	no interaction	no interaction	very low	-	-	-	-	-
Nimesulide	1,173,078	not mentioned	no interaction	no interaction	very low	-		-	-	-
Allopurinol	1,158,673	not mentioned	no interaction	no interaction	very low	-	-	-	-	-
Methylprednisolone	1,072,023	not mentioned	no interaction	no interaction	very low	-	-	-	-	-
Diosmin, Combination	1,041,036	not mentioned	not mentioned	not mentioned (no interaction)	-	-	-	-	-	-
Bisoprolol	988,934	not mentioned	no interaction	no interaction	very low	-	-	-	-	-
Metoprolol	951,519	not mentioned	no interaction	no interaction	very low	-	-	-	-	-
Desloratadine	928,466	not mentioned	no interaction	not mentioned (no interaction)	-	-	-	-	-	in the LC, only the risk of a potentially weak DI with loratadine in the sense of increasing its concentration for CYP 3A4 inhibition is mentioned
Pantoprazole	914,885	not mentioned	no interaction	no interaction	very low	-		-	-	-
Perindopril	912,726	not mentioned	no interaction	no interaction	very low	-	-	-	-	-
Tramadol and Paracetamol	891,294	not mentioned	potential interaction (TRA)	potential interaction	very low	possible higher risk of AE or lower effectiveness of tramadol due to its limited transformation to an active metabolite	-	dosage might have to be adjusted according to need	-	-
Other antibiotics for local application	887,182	not mentioned	not mentioned	not mentioned (no interaction)	-	-	-	-	-	due to the method of administration, the risk of systemic DI is very low
Cholecalciferol	847,158	not mentioned	no interaction	no interaction	very low	-	-	-	-	-
Amlodipine	729,801	possible interaction	potential interaction	potential interaction	very low	increase in the level of amlodipine due to inhibition of CYP 3A4 and P-gp	careful monitoring of therapeutic and adverse effects	recommended halving the dose of amlodipine or administration every other day (due to its long biological half-life) with resumption 3 days after the end of treatment; an alternative is to stop treatment under careful blood pressure monitoring	reduce the dose to 5 mg	-
Ramipril	666,829	not mentioned	no interaction	no interaction	very low	-	-	-	-	-
Telmisartan	657,758	not mentioned	no interaction	no interaction	very low	-	-	-	-	-
Levocetirizine	656,017	not mentioned	no interaction	not mentioned (no interaction)	-	-	-	-	-	according to LC, racemic cetirizine has no risk of DI
Escitalopram	655,028	not mentioned	no interaction	no interaction	very low	-	-	-	-	-
Mometasone	650,375	not mentioned	no interaction	no interaction	very low	-	-	-	-	-
Alprazolame	639,710	possible interaction	potential interaction	potential interaction	very low	possible increase in the level of alprazolam due to inhibition of its metabolism at the level of CYP 3A4	caution when administering	increased caution recommended during the treatment and for 3 days after the end of the treatment	reduce the dose by 50%	-
Furosemide	620,861	not mentioned	no interaction	no interaction	very low	-	-	-	-	-
Calcium, Combination with vitamin D and/or other medicine	613,597	not mentioned	no interaction	no interaction	very low	-	-	-	-	-
Clarithromycin	587,289	possible interaction	potential interaction	potential interaction	very low	possible increase in the level of clarithromycin due to inhibition of its metabolism at the level of CYP 3A4	there is usually no need to adjust the dose in people with normal kidney function	increased caution recommended during the treatment and for 3 days after the end of the treatment	-	-
Cefuroxime	585,459	not mentioned	no interaction	no interaction	very low	-	-	-	-	-
Sertraline	562,008	possible interaction	no interaction	potential interaction	very low	possible increase in serotonin levels due to inhibition of CYP 2D6	careful monitoring of therapeutic and adverse effects recommended	-	-	-
Perindopril, Amlodipine and Indapamide	561,772	possible interaction	potential interaction	potential interaction	very low	possible increase in both amlodipine and indapamide levels due to inhibition of its metabolism at the level of CYP 3A4	careful monitoring of therapeutic and adverse effects recommended	if necessary, reduce the dose (amlodipine) during the treatment and for 3 days after the end of the treatment	reduce the dose to 5 mg daily (amlodipine)	according to LC in relation to indapamide and amlodipine; according to SPC in relation only to amlodipine
Perindopril and Diuretics	560,088	not mentioned	potential interaction	potential interaction	very low	possible increase in the level of indapamide due to inhibition of its metabolism at the level of CYP 3A4	-	dose adjustment is usually not necessary; advice of increased caution	-	according to LC in relation to indapamide
Salbutamole	550,415	not mentioned	no interaction	no interaction	very low	-	-	-	-	-
Pitophenone and Analgesics	548,497	not mentioned	not mentioned	not mentioned (no interaction)	-	-	-	-	-	-
Dexamethasone and Anti-Infectives	533,175	possible interaction	potential interaction	potential interaction	very low	possible increase in the level of dexamethasone due to inhibition of its metabolism at the level of CYP 3A4	careful monitoring of therapeutic and adverse effects recommended	reduction in the dose by half recommended during treatment with resumption of the original dose 3 days after the treatment	-	according to LC, it only applies to doses of dexamethasone above 16 mg
Dexamethasone	518,989	possible interaction	potential interaction	potential interaction	very low	possible increase in the level of dexamethasone due to inhibition of its metabolism at the level of CYP 3A4	careful monitoring of therapeutic and adverse effects recommended	reduction in the dose by half recommended during treatment with resumption of the original dose 3 days after the treatment	-	according to LC, it only applies to doses of dexamethasone above 16 mg
Perindopril and Amlodipine	507,069	possible interaction	potential interaction	potential interaction	very low	possible increase in the level of amlodipine due to inhibition of its metabolism at the level of CYP 3A4	careful monitoring of therapeutic and adverse effects recommended	if necessary, reduce the dose (amlodipine) during the treatment and for 3 days after the end of the treatment	reduce the dose to 5 mg daily (amlodipine)	according to LC in relation to amlodipine
Azithromycin	467,662	not mentioned	no interaction	no interaction	very low	-	-	-	-	-
Influenza, Inactivated vaccine, Split virus or surface antigen	467,396	not mentioned	not mentioned	not mentioned (no interaction)	-	-	-	-	-	-
Trazodone	452,748	not mentioned	potential interaction	potential interaction	very low	possible increase in the level of trazodone due to inhibition of its metabolism at the level of CYP 3A4 with the risk of nausea, syncope or hypotension	-	as necessary, reduce the dose of trazodone during treatment and for 3 days after the end of the course	-	-
Prednisone	440,637	not mentioned	no interaction	no interaction	very low	-	-	-	-	-
Potassium chloride	434,269	not mentioned	no interaction	no interaction	very low	-	-	-	-	only potassium mentioned in LC
Tetanus toxoid	431,369	not mentioned	not mentioned	not mentioned (no interaction)	-	-	-	-	-	-
Cetirizine	424,934	not mentioned	no interaction	no interaction	very low	-	-	-	-	-
Tamsulosin	424,266	not mentioned	potential interaction	potential interaction	very low	possible increase in the level of tamsulosin due to inhibition of its metabolism at the level of CYP 3A4	-	interruption of tamsulosin treatment with resumption of the original dose 3 days after the treatment or reduction in its dose to 0.4 mg/day or administration every other day	suspend the use	-
Metamizole sodium	418,688	not mentioned	potential interaction	potential interaction	very low	possible increase in metamizole level by CYP inhibition and decrease in Paxlovid level by induction of CYP 3A4 and 2B6 by metamizole	-	-	-	-

* Total number of prescriptions of the given ATC group for the period September 2019–February 2022 in patients with COVID-19. 
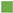
 without interaction; 
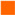
 potential interaction; 
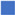
 not mentioned (without interaction).

**Table 2 pathogens-13-01055-t002:** A–D presents comparable data pertaining to the specialties of diabetology, infectious diseases, internal medicine, and general medicine. The information focuses on the top five prescribed medicinal substances within each of these specified disciplines.

A. Diabetology										
ATC Group	Number of Prescriptions Per Doctor *	SPC	Liverpool Checker (LC)	Total Risk	Evidence	Mechanism of Interaction	Recommended Measures According to SPC	Possible Measures According to LC	Recommendation of the Czech Society of Infectious Diseases	Note
Insulin glargine	175	not mentioned	no interaction	no interaction	very low	-	-	-	-	only “insulin” listed in LC
Insulin aspart	172	not mentioned	no interaction	no interaction	very low	-	-	-	-	only “insulin” listed in LC
Glimepiride	102	not mentioned	no interaction	no interaction	-	-	-	-	-	-
Gliclazide	99	not mentioned	no interaction	no interaction	-	-	-	-	-	-
Insulin degludek	83	not mentioned	no interaction	no interaction	-	-	-	-	-	-
**B. Infectious Diseases**										
**ATC Group**	**Number of Pescriptions Per Doctor ***	**SPC**	**Liverpool Checker (LC)**	**Total risk**	**Evidence**	**Mechanism of Interaction**	**Recommended Measures According to SPC**	**Possible Measures According to LC**	**Recommendation of the Czech Society of Infectious Diseases**	**Note**
Ceftriaxone	477	not mentioned	no interaction	no interaction	very low	-	-	-	-	-
Cefotaxime	127	not mentioned	no interaction	no interaction	very low	-	-	-	-	-
Vankomycin	69	not mentioned	no interaction	no interaction	very low	-	-	-	-	-
Meropenem	68	not mentioned	no interaction	no interaction	very low	-	-	-	-	-
Procaine benzylpenicillin	58	not mentioned	no interaction	no interaction	very low	-	-	-	-	penicillin as a whole listed in LC
**C. Internal medicine**										
**ATC Group**	**Number of Prescriptions Per Doctor ***	**SPC**	**Liverpool Checker (LC)**	**Total Risk**	**Evidence**	**Mechanism of Interaction**	**Recommended Measures According to SPC**	**Possible Measures According to LC**	**Recommendation of the Czech Society of Infectious Diseases**	**Note**
Food for special medical purposes (Czech ATV group)	26	not mentioned	not mentioned	not mentioned (no interaction)	-	-	-	-	-	-
Rivaroxabam	26	potential interaction	contraindication	contraindication	very low	possible increase in the level of rivaroxaban due to inhibition of CYP 3A4	simultaneous administration not recommended	the need to use alternative anticoagulant treatment	CI: use different antiviral drug	-
Apixaban	23	not mentioned	potential interaction	potential interaction	very low	possible increase in the level of rivaroxaban due to inhibition of CYP 3A4 and P-gp	-	consider transferring the patient to LMWH or acetylsalicylic acid with renewal 3 days after the end of treatment	CI: use different antiviral drug	-
Magnesium (combination of different salts)	19	not mentioned	no interaction	no interaction	-	-	-	-	-	-
Fenofibrate	19	not mentioned	no interaction	no interaction	-	-	-	-	-	-
**D. General medicine**										
**ATC Group**	**Number of Prescriptions Per Doctor ***	**SPC**	**Liverpool Checker (LC)**	**Total Risk**	**Evidence**	**Mechanism of Interaction**	**Recommended Measures According to SPC**	**Possible Measures According to LC**	**Recommendation of the Czech Society of Infectious Diseases**	**Note**
Warfarin	53	potential interaction	potential interaction	potential interaction	very low	possible changes in warfarin levels by inhibition of CYP 3A4 and induction of 1A2	monitoring of coagulation parameters	more careful INR control	-	-
Codeine	53	not mentioned	potentially weak interaction	potentially weak interaction	very low	possible reduction in codeine effectiveness due to inhibition of CYP 2D6	-	-	not suitable for combination with strong CYP 3A4 inhibitors	-
Indapamide	49	not mentioned	potential interaction	potential interaction	very low	possible increase in the level of indapamide due to inhibition of its metabolism at the level of CYP 3A4	-	more careful blood pressure control	-	-
Nebivolol	46	not mentioned	no interaction	no interaction	-	-	-	-	-	-
Losartan	44	not mentioned	potentially weak interaction	potentially weak interaction	very low	possible higher conversion of losartan to active metabolite due to induction of CYP 2C9	-	-	-	-

* The average number of prescriptions for patients with COVID-19 per doctor in the outpatient segment within the specified specialty during the observed period (September 2019–February 2022). red—contraindication, orange—potential interaction, blue—not mentioned (without interaction), green—without interaction, yellow—potentially weak interaction.

## Data Availability

Availability upon request from the main author.

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
