# Peer review of "Evaluating Drug Interaction Risks: Nirmatrelvir & Ritonavir Combination (PAXLOVID®) with Concomitant Medications in Real-World Clinical Settings"

_pathogens, 2024, doi:10.3390/pathogens13121055_

Round 1
Reviewer 1 Report
Comments and Suggestions for Authors
The article is devoted to the analysis of the efficacy and safety of combinations of various antiviral drugs used in the treatment of COVID-19 in the Czech Republic. At the same time, the focus of the work was the analysis of the potential interaction of Paxlovid with other commonly prescribed drugs in the treatment of COVID-19. At the same time, Paxlovid contains two antiviral drugs (nirmatrelvir and ritonavir). Paxlovid is currently in Phase III clinical trials for the treatment of COVID-19. The authors concluded that Paxlovid is a valuable option for the early treatment of COVID-19, but requires careful consideration of potential drug interactions, especially in patients with additional risk factors for COVID-19 complications, including the need for some patients to use immunosuppressants. In general, the article expands existing ideas about promising areas of COVID-19 therapy and may be useful to readers of the journal Pathogens. However, the authors should be aware of the following aspects of their work:
(1) Tables 1 and 2 have small and vague text and are poorly understood. They appear to be photographs from another paper by the authors.
(2) References need to be adapted to the MDPI style.
Author Response
Reviewer 1
The article is devoted to the analysis of the efficacy and safety of combinations of various antiviral drugs used in the treatment of COVID-19 in the Czech Republic. At the same time, the focus of the work was the analysis of the potential interaction of Paxlovid with other commonly prescribed drugs in the treatment of COVID-19. At the same time, Paxlovid contains two antiviral drugs (nirmatrelvir and ritonavir). Paxlovid is currently in Phase III clinical trials for the treatment of COVID-19. The authors concluded that Paxlovid is a valuable option for the early treatment of COVID-19, but requires careful consideration of potential drug interactions, especially in patients with additional risk factors for COVID-19 complications, including the need for some patients to use immunosuppressants. In general, the article expands existing ideas about promising areas of COVID-19 therapy and may be useful to readers of the journal Pathogens. However, the authors should be aware of the following aspects of their work:
- Tables 1 and 2 have small and vague text and are poorly understood. They appear to be photographs from another paper by the authors.
Thank you for your input. Due to the size of the Excel table, it was divided into sections, and a few screenshots were inserted into the Word document. The complete Excel table has been uploaded as part of the manuscript. I also confirm that the images are not photographs sourced from another paper.
- References need to be adapted to the MDPI style.
The references were updated according to MDPI style.

Reviewer 2 Report
Comments and Suggestions for Authors
In pathogens-3325055, Zatovkaňuková et al evaluate the drug-drug interaction risk between PAXLOVID and Concomitant Medications. The topic of the study is interesting and clinically relevant.
(1) The DDI caused by PAXLOVID is mainly due to CYP3A4. However, protein binding can also be another cause of DDI. What is the protein binding of the active substances in PAXLOVID?
(2) The polymorphism in CYP can also enhance the risk of DDI. The authors should also discuss this point and analyze the potential impact of polymorphism in CYP3A4 on DDI.
(3) Any information on hepatic / renal impairment on the metabolism of PAXLOVID. Will it enhance the risk of DDI?
Author Response
In pathogens-3325055, Zatovkaňuková et al evaluate the drug-drug interaction risk between PAXLOVID and Concomitant Medications. The topic of the study is interesting and clinically relevant.
(1) The DDI caused by PAXLOVID is mainly due to CYP3A4. However, protein binding can also be another cause of DDI. What is the protein binding of the active substances in PAXLOVID?
Pharmacokinetic interactions can naturally have different backgrounds. However, in the case of the FDC (nirmatrelvir + ritonavir) discussed here, interactions at the level of the cytochrome P450 play a dominant role. At the same time, the described interactions, which are reported in various sources (Liverpool, etc.), do not distinguish them a priori according to their nature, but according to their clinical significance, i.e. all currently known possible drug interactions are included in these sources.
Regarding the binding of the two active pharmaceutical ingredients, the following applies:
Nirmatrelvir :
There are species differences in plasma protein binding of nirmatrelvir. These are primarily driven by molecular differences in albumin and alpha-1 acid glycoprotein resulting in differences in binding affinity. Nirmatrelvir demonstrated moderate plasma protein binding in rats, monkeys, and humans with mean unbound fractions ranging from 0.310 to 0.478 (1).
According to the information given by the manufacturer, the protein binding of nirmatrelvir in human plasma is approximately 69% (2)
Ritonavir:
Plasma protein binding of ritonavir is high (96-99.5%) in tested animal species (male and female Sprague-Dawley rats, beagle dogs) and was nonsaturable in humans (HIV-negative male volunteers) at concentrations up to 30 micrograms/ml. Partitioning into the formed elements of whole blood was minimal (3).
According to the information given by the manufacturer, the protein binding of ritonavir in human plasma is approximately 98-99% (2).
The text was updated with the information on the page 3. Changes on the page are highlighted.
- The polymorphism in CYP can also enhance the risk of DDI. The authors should also discuss this point and analyze the potential impact of polymorphism in CYP3A4 on DDI.
Mutations in the CYP3A4 gene might lead to abolished, reduced, altered or increased enzymatic activity. In the literature, CYP3A4 polymorphism divides the general population into three groups – poor metabolizers, normal metabolizers, and rapid metabolizers, based on intronic single nucleotide polymorphism (SNPs) that modify expression levels rather than structure (4).
CYP3A4 is in the majority of individuals abundantly expressed in liver but population variability is extremely high (>100-fold), although complete absence of expression has not been definitively proven
The analysed prescriptions related to target population did not include a pre-defined population of CYP 3A polymorphisms. It can therefore be viewed as a population with a natural distribution of activity of this system according to the Gaussian curve. The vast majority of subjects will be extensively metabolisers – Figure 1 (4).
Figure 1: Sparteine oxidation phenotype and genotype distribution in a German population (4)
However, the clinical significance of possible polymorphisms in relation to the medication administered on this isoform is usually mentioned as significantly lower compared to, for example, the highly polymorphic isoform 2D6 or 2C19 (4-8).
The text was updated accordingly on the page 9. The changes are highlighted in the updated manuscript.
(3) Any information on hepatic / renal impairment on the metabolism of PAXLOVID. Will it enhance the risk of DDI?
Compared to healthy controls with no renal impairment, the Cmax and AUC of nirmatrelvir in patients with mild renal impairment was 30% and 24% higher, in patients with moderate renal impairment was 38% and 87% higher, and in patients with severe renal impairment was 48% and 204% higher, respectively (2).
Compared to healthy controls with no hepatic impairment, the pharmacokinetics of nirmatrelvir in participants with moderate hepatic impairment was not significantly different. Adjusted geometric mean ratio (90% CI) of AUCinf and Cmax of nirmatrelvir comparing moderate hepatic impairment (test) to normal hepatic function (reference) was 98.78% (70.65%, 138.12%) and 101.96% (74.20%, 140.11%), respectively (2).
According to the product information, Paxlovid is contraindicated in patients with severe hepatic or renal impairment. No dose adjustment is required for patients with mild (Child-Pugh Class A) or moderate (Child-Pugh Class B) hepatic impairment, or for those with mild renal impairment (eGFR ≥ 60 to < 90 mL/min). However, in patients with moderate renal impairment (eGFR ≥ 30 to < 60 mL/min), the dose should be reduced to nirmatrelvir/ritonavir 150 mg/100 mg every 12 hours for 5 days to prevent over-exposure (2).
The text was updated accordingly on page 10. The changes are highlighted in the updated manuscript.
